# ROYAL SOCIETY
# OPEN SCIENCE

# Research

artificial intelligence/pattern recognition/statistics

topic modelling, matrix factorization, temporal/dynamic methods, crime analysis

**Author for correspondence:**
Seppo Virtanen
e-mail: s.virtanen@leeds.ac.uk

# Uncovering dynamic textual topics that explain crime

Seppo Virtanen

School of Mathematics, University of Leeds, Leeds LS2 9JT, UK

SV, 0000-0001-9556-1369

Crime analysis/mapping techniques have been developed and applied for crime detection and prevention to predict where and when crime occurs, leveraging historical crime records over a spatial area and covariates for the spatial domain. Some of these techniques may provide insights for understanding crime and disorder, especially, via interpreting the weights for the spatial covariates based on regression modelling. However, to date, the use of temporal covariates for the time domain has not played a significant role in the analysis. In this work, we collect time-stamped crime-related news articles, infer crime topics or themes based on the collection and associate the topics with the historical numeric crime counts. We provide a proof-of-concept study, where instead of adopting spatial covariates, we focus on temporal (or dynamic) covariates and assess their utility. We present a novel joint model tailored for the crime articles and counts such that the temporal covariates (latent variables, more generally) are inferred based on the data sources. We apply the model for violent crime in London.

## 1. Introduction

Textual streaming news articles from reputable sources provide easily accessible real-time detailed information not only about significant and prominent crime events, that affect society but also insights and analyses exploring crime trends and causes and effects of crime on society. The articles combine several different information channels such as criminologists, local councils, criminal proceedings and government, for instance, in addition to the police. The articles may cover events involving gun/knife crime, rioting, violent demonstrations, sexual assaults, abuse, domestic violence, terrorism and hate crime, for instance. Each article may detail information regarding offender(s), victim(s), location, time, motive and explanation of the crime. In addition, some articles may cover crime trends and summaries and government/police responses.

On the other hand, numeric crime counts based on reported crime are being provided by the police online. Each offence includes a discrete time stamp, region and crime category that

are suitably anonymized for confidentiality. Analysis of the widely accessible crime counts forms the backbone of criminology inferring plausible theories and patterns that explain crime. Accordingly, several methods for analysing the count data have been presented and deployed for crime prediction [1,2].

We hypothesize that the *textual* time-stamped crime news articles provide a rich source of information and context that may be used to explain and predict *numeric* crime counts. Informally, our intuition is to use the textual data to explain inferred patterns of criminal activity based on the counts. We avoid a labour-intensive manual approach to search for matching/underlying text content and automate the approach using statistical modelling in the context of explainable artificial intelligence or machine learning. Such analysis is of key interest, for instance, for detecting and preventing crime, assessing policing effects, understanding policing demand and criminal activity as well as uncovering non-trivial (dynamic) associations between news coverage and crime. The text data complements numeric official records covering information regarding the offender and victim and the type/details of offence that are important for understanding underlying causes and mechanisms of crime.

However, the textual and numeric crime data streams are not in general paired one-to-one; one major offence may attract multiple news articles, most (minor) offences fail to appear in the news because of low relevance, and some news articles that report on crime trends and summaries are based on several offences. We propose to couple the data sources based on non-overlapping consecutive time windows/frames such that we collect all news articles and reported crime counts during each time window. Based on the temporally coupled datasets, our aim is to uncover dynamic patterns or themes that explain crime.

We present a statistical joint model for the data sources combining dynamic topic modelling and Poisson matrix factorization by suitably sharing latent variables between the separate models/data sources across the time stamps/windows. Topic models are ubiquitous probabilistic models for text documents that enable exploring and summarizing large text collections [3,4], such as news articles, in a meaningful manner. In general, the topics are deemed to correspond to certain meaningful and useful textual themes. Dynamic topic models are suitable for documents that are grouped according to time stamps/windows, as considered in this work, capturing topic evolution and temporal trends [5]. Dynamic Poisson matrix factorization models are suitable for analysis of sequential count data collected in a matrix whose rows (and/or columns) exhibit temporal dependence [6,7]. Here, numeric crime counts may naturally be represented in a matrix where columns correspond to regions and rows to time stamps such that each element of the matrix contains a count for a specific crime category for each region and time stamp.

The proposed model may be interpreted as a novel supervised dynamic topic model motivated by the prediction task. The model goes beyond standard supervised topic models [8–11] that assume each document is paired with one or more response variables (one-to-one mapping) and hence are not as such appropriate for the task considered. The models infer topics that are predictive of the response variables. For our model, the response variables correspond to the dynamic crime counts.

The proposed joint model performs often better, as verified in the experiments, when compared with an alternative two-step process that, firstly, infers the topics or, more generally, (manually) collects dynamic covariates and, secondly, performs a separate Poisson regression of the counts based on the covariates finding which topics/text content are associated with crime counts. Whereas traditional regression models require the covariates to be observed and fixed, our approach is to infer them based on the observed data, automating the process of constructing covariates. From a statistical perspective, the joint model is able to take uncertainties properly into account and leverage potentially weakly shared information between the text and counts, contrarily to the two-step modelling approach.

We note that not all free-form text content is associated with or predictive of crime counts. We propose to include a latent mechanism for the model to explain away text content that is not relevant for the counts. Words that are not explained by the topics are explained by the alternative mechanism improving predictive performance and inferring more useful and meaningful topics that focus on explaining crime.

It is widely acknowledged that crime is clustered in both space and time, explaining bursts of criminal activity following so-called near-repeat pattern theory [12]. The clustering property of crime translates to elevated risk of crime for certain areas (so-called, hot spots) following different temporal rates of spread and decay. Evidently, the clustering property is also seen in the crime news. In addition, news content itself tends to cover for some time information of particular events or trends. Our model is able to leverage the inherent clustering property and impose dynamics in a flexible manner. Topics associated with events may show different dynamics from topics that explain overall trends or patters of crime. The topic timelines showing topic activations over time may be bursty and sparse or slowly varying with a slow increase/decrease or take a constant value. We depart from existing dynamic topic models [5], that assume similar dynamics for all topics, and present a flexible dynamic Gamma

process for the (positively valued) topic time series, extending the process by Virtanen & Girolami [13] introducing topic-specific dynamics suitable for the application. We emphasize that both data sources affect inference of the topic dynamics/timelines.

Offence categorization by the police is not able to account for offence details that would provide insight for multi-faceted and complex crime. Categorization may also be subject to ambiguity. For example, violent crime, as considered in this work, ranges from murder and grievous bodily harm to common assault and harassment. We note that different types of offences may exhibit complicate spatio-temporal inter-relationships, following criminological theories, such as repeat offending and victimization, clustering property and cycle of crime. Severe but infrequent offences have high impact and less severe but frequent offences affect local communities. Analysis of severe crime may suffer from the data scarcity problem, but analysis of both severe and less severe crime may be confounded by the large counts for less severe crime. We exploit total crime counts over a high-level offence category and leverage the text data to infer topics that provide important insights regarding the type and severity of offences providing a complete summary of crime without suffering from the data scarcity problem.

Offence-level textual police reports, that may be brief, repetitive and rely on a specific vocabulary including codes and abbreviations, remain confidential with limited accessibility for (manually) moderated content. Hence, we propose to collect and analyse accessible high-quality crime news articles that provide similar and more general information about crime. The quality of topics naturally depends primarily on the quality of the text data. Major factors affecting the quality include scale, scope, style and content of the articles. For these factors, we use full news articles provided by a reputable newsagent instead of brief text snippets capturing 'breaking news' or user-generated content for social media such as tweets.

Because of confidentiality, some reported offences may not be accounted for in the crime count data. Also, not all crime is reported to the police in the first place, possibly because of sensitivity issues or low expected utility of doing so. Such missing data complicates detection/inference of the crime patterns. Our aim is to alleviate this problem by combining information from the news articles. On the other hand, the missing data problem potentially undermines quantitative evaluation of predictive performance of the models. It is important to complement evaluation by carrying out detailed qualitative analysis as well by interpreting the model.

The paper is structured as follows. Section 2 introduces the application and data. We provide a proof-of-concept study for analysing violent urban crime. The proposed joint model for a dynamic collection of crime news articles and counts is presented in §3. Section 4 shows both quantitative and qualitative results. We carry out a model comparison assessing predictive performance and interpret the inferred topics and their corresponding timelines. We show that the topics capture a wide range of useful information about criminal activity/behaviour, crime events/trends as well as police, court and government actions. Section 5 concludes the paper with a discussion.

## 2. Data

In this work, we study violent urban crime, that has significant media coverage and impact for the society, occurring in London, a major metropolitan area, that experiences high crime rates. We collect monthly crime counts for London boroughs provided by the Metropolitan police force from January 2008 to December 2018 for a crime category of violence against the person.[1] The total number of crime offences is $2.12 \times 10^6$ and on average each month contains $66 \times 10^3$ offences over $M = 32$ boroughs.

For each month, we combine the crime counts with crime-related news articles provided by the Guardian,[2] published during that time interval. The Guardian is a highly reputable newspaper that follows a formal style with a global audience. We use the Guardian API specifying tags given by crime and knife crime with a keyword London to search for articles appearing in the UK or world news sections. For controlled quality, we discard articles that correspond to opinions, comments or other content provided by users/readers.

We adopt standard text data processing removing non-text characters, such as numbers and punctuation, and stop words.[3] We keep news articles that contain more than (including) 10 words

---

[1]https://data.london.gov.uk/dataset/recorded_crime_summary#.

[2]https://open-platform.theguardian.com/.

[3]We use the R function *stopwords(en)* of the package stopwords [14].

and remove words that appear in only one article. The data contain $T = 132$ time points (months) and 3289 news articles. The vocabulary contains $V = 24\,115$ words and the number of words over the collection is $1.15 \times 10^6$. On average, each month contains 25 articles and each article contains 351 words.

The crime count for the $m$th region and $t$th time stamp is denoted by $y_{t,m}$. Here, $m$ takes values over $M$ regions. The $i$th article for the $t$th time stamp is denoted by $\boldsymbol{w}_{t,i} = \{w_{t,i}^{(1)}, \ldots, w_{t,i}^{(N_{t,i})}\}$, where $w_{t,i}^{(n)}$, for $n = 1, \ldots, N_{t,i}$, denote individual words taking values over the word vocabulary (mathematically equivalently represented as discrete indices, $w_{t,i}^{(n)} \in \{1, \ldots, V\}$).

# 3. Model

Our model infers a set of topics that are distributions over the word vocabulary. The elements of the distribution are positive and sum to one over the vocabulary. The topics are deemed to capture meaningful themes that may be inspected based on top-probability words of the inferred topics. Examples of inferred topics based on the news articles with top words can be found in the corresponding tables of figures 3–8.

Each article is expressed as a set/bag of words and the model assumes the words are generated from a categorical distribution, whose expectation parameters correspond to topics. Each article contains a distribution (proportion) over the topics indicating which topics, and to what degree, are relevant for each article. These topic proportions are assumed to be generated from a Dirichlet distribution.

We group articles according to time stamps or windows and assume a dynamic process for the topic proportions to capture sequential dependence between the time stamps corresponding to topic evolution and temporal trends. The model assumes separate Dirichlet parameters for each time-stamped group that follow a Markov-type dependence assumption. We note that a similar construction may be used also for the topics but we do not explore that variant further in this work because of increased computational demand.

Numeric crime counts are collected in a matrix over time stamps/windows and regions. The text and count datasets are coupled over the time windows such that each group of news articles co-occurs with corresponding rows of the count matrix.

We assume the counts are generated from Poisson distributions. The model contains a set of latent variables for each time window and the expectation parameters of the Poisson distributions for each time window and region are given by weighted linear combinations of the latent variables. For the combination, we normalize the positively valued latent variables to prevent scale ambiguity. Correspondingly, the positive weights, that are introduced for each region, may be directly interpreted as expected crime counts.

Our joint model of crime news articles and counts assumes the latent variables correspond to the parameters of the Dirichlet distributions for the topic proportions for each group/time window. The latent variables are shared between the crime news and counts. Each topic is coupled with the weights, permitting meaningful and intuitive model interpretation. Key model variables for inspection include the latent variables, topics and weights.

Figure 1 shows a graphical illustration of our model. We denote the set of $K$ latent variables as $\alpha_{t,k}$, for $t = 1, \ldots, T$ and $k = 1, \ldots, K$. The latent variables indicate which topics (thematic word distributions over the vocabulary) denoted by $\boldsymbol{\eta}_k$, for $k = 1, \ldots, K$, are associated with and to what degree for each group or time stamp. An article-specific topic proportion (a probability distribution) over the $K$ topics is generated from a Dirichlet distribution

$$\boldsymbol{\theta}_{t,i} \sim \text{Dirichlet}(\boldsymbol{\alpha}_t), \tag{3.1}$$

for $i = 1, \ldots, N_t$, where $N_t$ denotes the number of articles for the $t$th time stamp. The latent variables represent expected topic proportions for each month, such that,

$$E[\boldsymbol{\theta}_{t,i}] \propto \boldsymbol{\alpha}_t,$$

for $t = 1, \ldots, T$. The scale of the values controls the variance of the distribution.

The crime counts follow Poisson distributions; we assume

$$y_{t,m} \sim \text{Poisson}(\rho_{t,m}), \tag{3.2}$$

for $t = 1, \ldots, T$ and $m = 1, \ldots, M$, with parameters denoted as $\rho_{t,m}$, respectively. We denote the weights for the $m$th region and $k$th topic as $\omega_{k,m}$. The parameters are given by linear combinations of the normalized

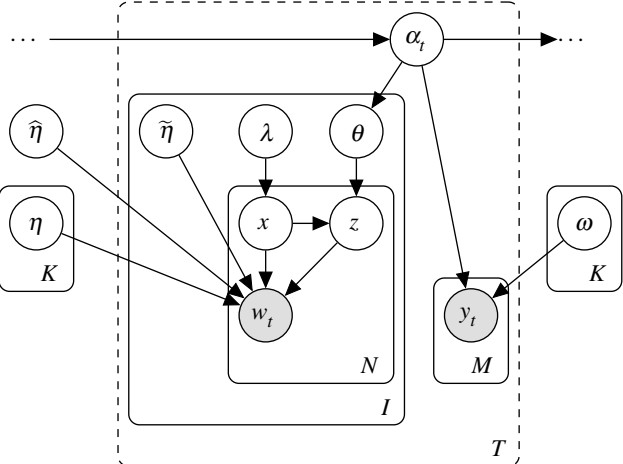

**Figure 1.** Graphical plate diagram of the model. The latent variables $\boldsymbol{\alpha}_t$ depend on $\boldsymbol{\alpha}_{t-1}$, affect $\boldsymbol{\alpha}_{t+1}$ and are shared between the dynamic topic model (nodes below the left edge of $\boldsymbol{\alpha}_t$) and Poisson matrix factorization model (nodes below the right edge of $\boldsymbol{\alpha}_t$). The temporal plate indexing $t$ is dashed to emphasize dynamic dependence.

latent variables and weights

$$\rho_{t,m} = \sum_{k=1}^{K} \frac{\alpha_{t,k}}{\sum_{k'=1}^{K} \alpha_{t,k'}} \omega_{k,m}. \tag{3.3}$$

The parameters capture the expected number of crime counts for each month and borough, $E[y_{t,m}] = \rho_{t,m}$. The weights may be directly interpreted as expected crime counts over the regions because of the normalization constraint.

We note that not all text content is related to crime counts and we introduce article-specific word distributions as well as a shared word distribution to explain away non-crime-related content not explained by the topics. We introduce word-specific switch variables $x$ to indicate whether the topics ($x = 0$), article-specific ($x = 1$) or common distributions ($x = 2$) explain it. For the topic route (first), we draw a topic assignment

$$z_{t,i}^{(n)} \sim \text{Categorical}(\boldsymbol{\theta}_{t,i}) \tag{3.4}$$

and condition on that value to draw the word from the corresponding topic

$$w_{t,i}^{(n)} \sim \text{Categorical}(\boldsymbol{\eta}_{z_{t,i}^{(n)}}). \tag{3.5}$$

For the second route, we draw the word from the article-specific distribution

$$w_{t,i}^{(n)} \sim \text{Categorical}(\widehat{\boldsymbol{\eta}}_{t,i}).$$

For the third route, we draw the word from the common word distribution

$$w_{t,i}^{(n)} \sim \text{Categorical}(\widetilde{\boldsymbol{\eta}}).$$

We repeat the process for all words for each article. The switch variable for each word is generated as

$$x_{t,i}^{(n)} \sim \text{Categorical}(\lambda_{t,i}\mathbf{1}_3),$$

where the expectation parameter equals over the three possible categories with an article-specific value $\lambda_{t,i}$. The generative model for the words builds on the non-dynamic topic model with fixed symmetric Dirichlet priors for the topics by Chemudugunta *et al.* [15]. Their motivation is to separate article-specific or common content from the topic content. However, our motivation is to improve the predictive performance of the joint model.

We assume the latent variables follow a dynamic Gamma process with topic-specific dynamics for increased modelling flexibility. We assume

$$\alpha_{t,k} \sim \text{Gamma}\left(\tau_k, \frac{\tau_k}{\alpha_{t-1,k}}\right),$$

where

$$\tau_k \sim \mathrm{Gamma}(\alpha_0^{(\tau)}, \beta_0^{(\tau)}),$$

for $t = 2, \ldots, T$ and $k = 1, \ldots, K$. For $t = 1$, we assume

$$\alpha_{1,k} \sim \mathrm{Gamma}(\alpha_0, \beta_0),$$

respectively. Here, the $\tau_k$ controls the dynamic properties of $\alpha_{t,k}$, for $t = 1, \ldots, T$; a large value leads to a smooth process with small fluctuations, whereas small values lead to a less smooth (bursty) process with potentially rapid and large fluctuations.

To complete the model construction, we assign symmetric Dirichlet distributions for the topics and word distributions with value $\gamma$. For the positive weights, we employ a sparsity-promoting prior distribution, $\omega_{k,m} \sim \mathrm{Exponential}(a)$. For $\lambda_{t,i}$, $a$, $\alpha_0^{(\tau)}$ and $\beta_0^{(\tau)}$, we assign weakly informative Gamma prior distributions. Accordingly, we also set weakly informative values for the hyperparameters $\gamma = 0.01$ and $\alpha_0 = \beta_0 = 1$.

For posterior inference, we use collapsed Gibbs sampling for the topic and route assignments analytically marginalizing out the topics and word distributions, following Chemudugunta *et al.* [15]. For the latent variables and the remaining variables, we adopt slice sampling, following Virtanen & Girolami [13].

# 4. Results

## 4.1. Quantitative model comparison

Our model has one parameter to validate; the number of topics. We perform model validation using predictive performance for the crime counts. We leave randomly out 20% of the crime counts and predict these values based on the model posterior distribution. We randomly generate five different missing value masks and average performance over them. For the $i$th mask, we evaluate log likelihood for the held-out observations, denoted by the index set $\mathcal{M}_i$,

$$\mathcal{L}_i = \frac{1}{S} \sum_{s=1}^{S} \sum_{m,t \in \mathcal{M}_i} \log \mathrm{Poisson}(y_{t,m} | \rho_{m,t}^{(s)}),$$

averaging over $S$ posterior samples. We also evaluate Watanabe–Akaike information criteria (WAIC) [16], defined as,

$$\mathrm{WAIC}_i = \frac{1}{S} \sum_{s=1}^{S} \sum_{m,t \notin \mathcal{M}_i} \log \mathrm{Poisson}(y_{t,m} | \rho_{m,t}^{(s)}) - \mathrm{Var}_i, \tag{4.1}$$

where $\mathrm{Var}_i$ denotes the variance of the likelihood values for observations over posterior samples. For WAIC (4.1), the higher the value, the better. Finally, we also report mean absolute error (MAE) values for held-out observations, defined as,

$$\mathrm{MAE}_i = \frac{1}{S} \sum_{s=1}^{S} \sum_{m,t \in \mathcal{M}_i} \frac{|y_{t,m} - \rho_{m,t}^{(s)}|}{\#[\mathcal{M}_i]},$$

where $\#[\cdot]$ computes the number of items in the input set. In total, we run the sampler for $2 \times 10^5$ iterations, that suffices for convergence based on standard convergence diagnostics, and retain the last $10^3$ samples to approximate the posterior distribution. We search for $K = \{100, 200, 300, 400\}$. Table 1 shows that the model performance does not significantly improve for $K \geq 300$. Thus we show results for $K = 300$ in the following.

We also compare against dynamic matrix factorization (MF) and Poisson regression (PR) models for the crime counts. The MF approach is a special case of our model; the crime counts are modelled using Poisson distributions, following equations (3.2) and (3.3). Importantly, the MF model uses only crime counts and hence the latent variables $\boldsymbol{\alpha}$ are not shared with the news articles. The PR approach is similar to the MF model; however, the latent variables $\boldsymbol{\alpha}$ (or covariates/inputs in this context) are fixed for the regression model. For the PR approach, we use covariates that consist of the inferred topic proportions of a text-based dynamic topic model, following equations (3.1), (3.4) and (3.5). This topic model is a special case of our model that uses only crime articles and the latent variables $\boldsymbol{\alpha}$ are

**Table 1.** Model selection for our model for different component numbers. The values show mean and s.d. for predictive log likelihood and mean absolute error (MAE) for held-out observations over the folds. Similarly, we also show values for WAIC but using in-sample observations.

| K | 100 | 200 | 300 | 400 |
|---|---|---|---|---|
| log likelihood | $-4509 \pm 35$ | $-4496 \pm 48$ | $-4480 \pm 48$ | $-4483 \pm 40$ |
| WAIC ($\times 10^3$) | $-16.70 \pm 0.12$ | $-16.72 \pm 0.091$ | $-16.72 \pm 0.097$ | $-16.84 \pm 0.15$ |
| MAE | $29.55 \pm 0.5$ | $29.27 \pm 0.71$ | $29.18 \pm 0.51$ | $29.04 \pm 0.47$ |

**Table 2.** Model comparison against MF and PR approaches. The values show mean and s.d. for predictive log likelihood and mean absolute error (MAE) for held-out observations over the folds. Similarly, we also show values for WAIC but using in-sample observations.

| model | our | MF | PR |
|---|---|---|---|
| log likelihood | $-4480 \pm 48$ | $-4454 \pm 75$ | $-5347 \pm 156$ |
| WAIC ($\times 10^3$) | $-16.76 \pm 0.11$ | $-16.68 \pm 0.09$ | $-113.21 \pm 9.49$ |
| MAE | $29.18 \pm 0.51$ | $29.03 \pm 0.51$ | $41.23 \pm 1.6$ |

not shared with the crime counts. The comparison avoids any potential method-specific bias and focuses on evaluating the effect of jointly modelling crime news articles and counts. Following similar experimental settings, table 2 shows that our model for $K = 300$ attains the same performance as MF and both models perform better than PR. The results show that the two-step approach is unable to infer meaningful topic timelines for crime count prediction and verifies that the topics and temporal trends differ between our joint model and the text-based model. Our model improves over MF by increased interpretability; the component timelines of the MF approach may not be easily interpretable. Figure 2 shows predictions versus observations for each method, for a particular mask and three representative regions, in order to prevent visual clutter. Based on the figure, we see that our model and MF perform equally well and much better than the PR approach. PR is only able to infer general trends discarding more fine-grained details. In the following, we carry out exploratory analysis inspecting topics that explain in a meaningful manner such crime trends as well as peaks and fluctuations as illustrated in figure 2.

## 4.2. Inspection of the inferred topics

In the following, we inspect inferred topics of our model in more detail, collecting topics into seven groups: a recent rise of violent crime, general violence, the London riots, terrorism, criminal justice, policing as well as reasons and causes of violent crime. The groups correspond to figures 3–8, respectively. For each group, we label the topics based on the corresponding figure number for clarity.

We show average posterior topic proportions (timelines) over the time range, $\alpha_{t,k}$, for $t = 1, \ldots, T$, and the associated most probable words based on $\eta_k$, for the selected subsets of topics (here, inspection of all the 300 topics is not feasible). The timelines show when and to what extent the topics contribute to explaining crime and the top words reveal themes or content of the topics. We also show the proportion of explained crime for each topic, based on topic-specific contributions to computing $\rho$ over time stamps and regions. These values summarize relevance of each topic for crime counts. Accordingly, for increased visual clarity, we normalize the topic timelines individually. We compute posterior averages using the posterior samples. We did not find label switching between latent variables and thus averaging over the samples is meaningful. We note that inspection of the spatial weights that capture the spatial distribution of expected crime counts over the study area is out of scope of this study. We summarize top boroughs with highest and lowest weights over the inspected topics. The top-5 boroughs most associated with violent crime include Westminster, Lambeth, Southwark, Newham and Ealing. Contrarily, boroughs with least crime include Richmond, Kingston upon Thames, Sutton, Merton and Havering.

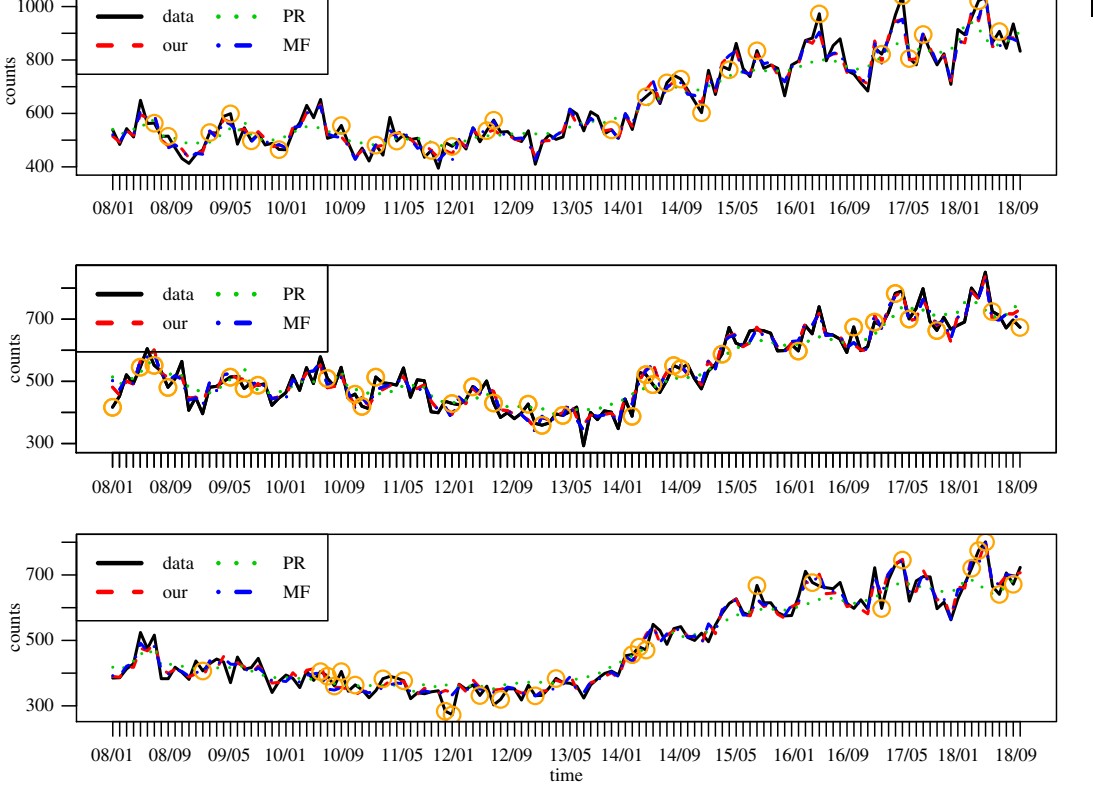

**Figure 2.** Crime count predictions for each method. The held-out values are indicated by orange squares. The top row is for Tower Hamlets, middle for Camden and bottom for Barnet.

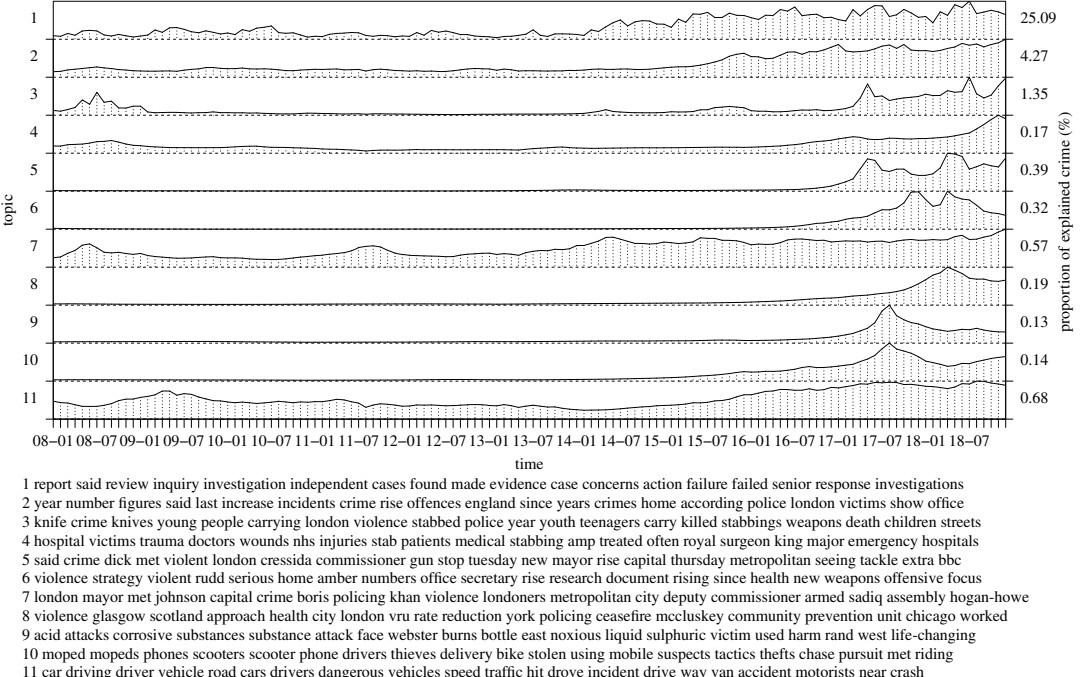

1 report said review inquiry investigation independent cases found made evidence case concerns action failure failed senior response investigations
2 year number figures said last increase incidents crime rise offences england since years crimes home according police london victims show office
3 knife crime knives young people carrying london violence stabbed police year youth teenagers carry killed stabbings weapons death children streets
4 hospital victims trauma doctors wounds nhs injuries stab patients medical stabbing amp treated often royal surgeon king major emergency hospitals
5 said crime dick met violent london cressida commissioner gun stop tuesday new mayor rise capital thursday metropolitan seeing tackle extra bbc
6 violence strategy violent rudd serious home amber numbers office secretary rise research document rising since health new weapons offensive focus
7 london mayor met johnson capital crime boris policing khan violence londoners metropolitan city deputy commissioner armed sadiq assembly hogan-howe
8 violence glasgow scotland approach health city london vru rate reduction york policing ceasefire mccluskey community prevention unit chicago worked
9 acid attacks corrosive substances substance attack face webster burns bottle east noxious liquid sulphuric victim used harm rand west life-changing
10 moped mopeds phones scooters scooter phone drivers thieves delivery bike stolen using mobile suspects tactics thefts chase pursuit met riding
11 car driving driver vehicle road cars drivers dangerous vehicles speed traffic hit drove incident drive way van accident motorists near crash

**Figure 3.** Recent rise of violent crime.

Figure 3 captures topics related to a trend of increasing violent crime towards the end of the time frame of the study. The topics show an increase in crime reporting (1.1 and 1.2), emergence of knife crime among youth (1.3) and associated hospitalizations (1.4). The following topics (1.5–1.8) involve

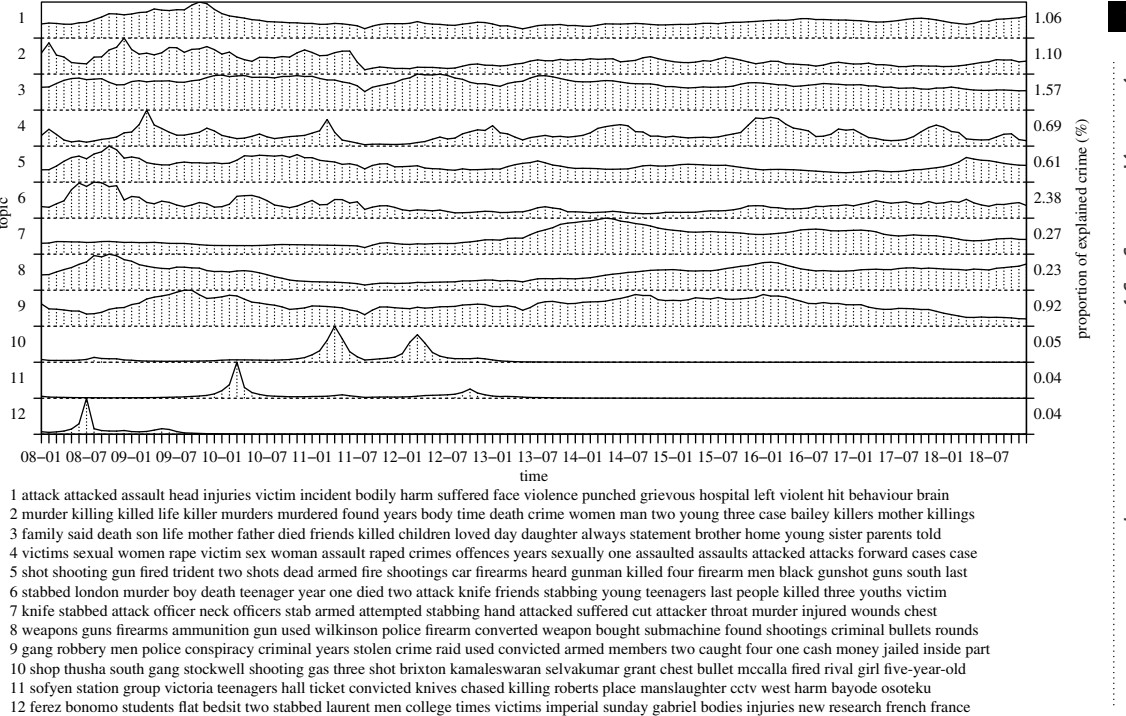

1 attack attacked assault head injuries victim incident bodily harm suffered face violence punched grievous hospital left violent hit behaviour brain
2 murder killing killed life killer murders murdered found years body time death crime women man two young three case bailey killers mother killings
3 family said death son life mother father died friends killed children loved day daughter always statement brother home young sister parents told
4 victims sexual women rape victim sex woman assault raped crimes offences years sexually one assaulted assaults attacked attacks forward cases case
5 shot shooting gun fired trident two shots dead armed fire shootings car firearms heard gunman killed four firearm men black gunshot guns south last
6 stabbed london murder boy death teenager year one died two attack knife friends stabbing young teenagers last people killed three youths victim
7 knife stabbed attack officer neck officers stab armed attempted stabbing hand attacked suffered cut attacker throat murder injured wounds chest
8 weapons guns firearms ammunition gun used wilkinson police firearm converted weapon bought submachine found shootings criminal bullets rounds
9 gang robbery men police conspiracy criminal years stolen crime raid used convicted armed members two caught four one cash money jailed inside part
10 shop thusha south gang stockwell shooting gas three shot brixton kamaleswaran selvakumar grant chest bullet mccalla fired rival girl five-year-old
11 sofyen station group victoria teenagers hall ticket convicted knives chased killing roberts place manslaughter cctv west harm bayode osoteku
12 ferez bonomo students flat bedsit two stabbed laurent men college times victims imperial sunday gabriel bodies injuries new research french france

**Figure 4.** Violence-related topics.

interventions or statements by government, police and commissioner to reduce and address the rise of violent crime. Other crime trends include acid attacks, moped thieves and dangerous driving that further evidence less secure streets, captured by topics (1.9–1.11), respectively. These topics and their underlying temporal trends may be useful for policing and resource allocation, identifying the type of problem to address and understanding demand. Most topics provide information regarding the offender, victim, and context and process of crime. The topic 1.1 explains a large proportion of crime, whereas the remaining topics contribute in smaller but roughly similar quantities. We also note that the dynamics between different topics vary significantly; some show complex and rapid fluctuations whereas some are more or less constant with slow or no temporal changes, supporting our model assumption of topic-specific dynamics. Some of the topics further show that the model is able to infer topics that are active only during a certain time window, corresponding to birth and death process of topics. In other words, topic timelines capture duration of elevated risk for certain type of crime as explained by the topics. Analysis of the rate of spread or decay may give further evidence for interventions, resource allocation as well as policy evaluation.

Figure 4 captures topics that are related to violent crime more generally. The topics include bodily harm (2.1), murder (2.2), death in family (2.3), sexual offences (2.4), shootings (2.5), stabbings (2.6 and 2.7), weapons (2.8) and (personal) robberies (2.9). Interestingly, most of these topics correspond directly to different subcategories (or types) of violent crime or crime including a threat of violence. The topics provide detailed information about the context of the offences (mechanism and cause) useful for policing and understanding the type and severity of the offences. The sexual offences topic (2.4) exhibits rapid temporal fluctuations corresponding to particular events whereas the remaining topics show more or less constant trends that contribute to crime counts uniformly over time. Here, the topics involving knives (2.6–2.7) differ from (1.3), that highlights youth carrying knives and youth knife crime in London, to serious stabbings (major events) towards youth and police, correspondingly. Many of the topics (all topics are not shown) are associated with particular events (or incidents). For example, Brixton shop shooting, Victoria station knife attack and New Cross double murder, are captured by topics 2.10–2.12, respectively, containing detailed information about victim(s), offender(s) and location(s) regarding names and more specific vocabulary. These events have minor overall contribution to explaining crime counts but provide useful insights, following routine activity and crime pattern theories, [17–19], respectively. The routine activity theory explains that each event/offence must include the presence of offender, victim and opportunity or lack of supervision to prevent/deter crime from taking place. The crime

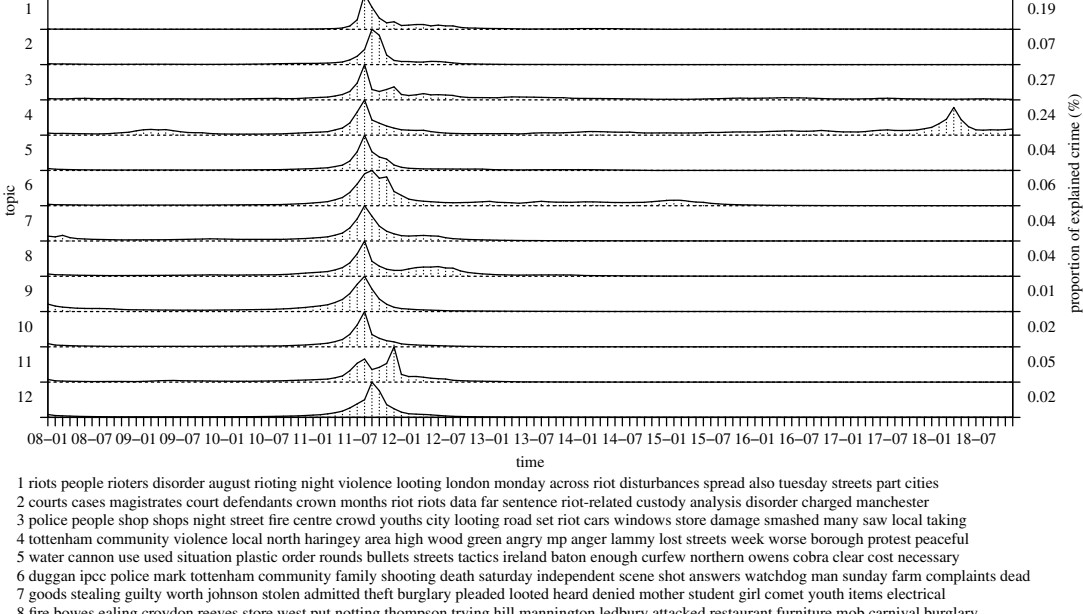

1 riots people rioters disorder august rioting night violence looting london monday across riot disturbances spread also tuesday streets part cities
2 courts cases magistrates court defendants crown months riot riots data far sentence riot-related custody analysis disorder charged manchester
3 police people shop shops night street fire centre crowd youths city looting road set riot cars windows store damage smashed many saw local taking
4 tottenham community violence local north haringey area high wood green angry mp anger lammy lost streets week worse borough protest peaceful
5 water cannon use used situation plastic order rounds bullets streets tactics ireland baton enough curfew northern owens cobra clear cost necessary
6 duggan ipcc police mark tottenham community family shooting death saturday independent scene shot answers watchdog man sunday farm complaints dead
7 goods stealing guilty worth johnson stolen admitted theft burglary pleaded looted heard denied mother student girl comet youth items electrical
8 fire bowes ealing croydon reeves store west put notting thompson trying hill mannington ledbury attacked restaurant furniture mob carnival burglary
9 help clean volunteers council hackney thompson brooms riotcleanup gloves debris traders mare clarence convenience councils tweeted mess shopkeepers
10 jahan men birmingham calm haroon brothers three deaths abdul shazad winson dudley ali reported west tariq protecting leave petrol musavir
11 like just police said everyone tv tottenham friends stuff blackberry yeah running got friend shops throwing hate ain van fire mostly watching
12 sentences facebook sentencing guidelines blackshaw advice justices clerks warrington committed sutcliffe-keenan men four consider jordan inciting

**Figure 5.** London riots.

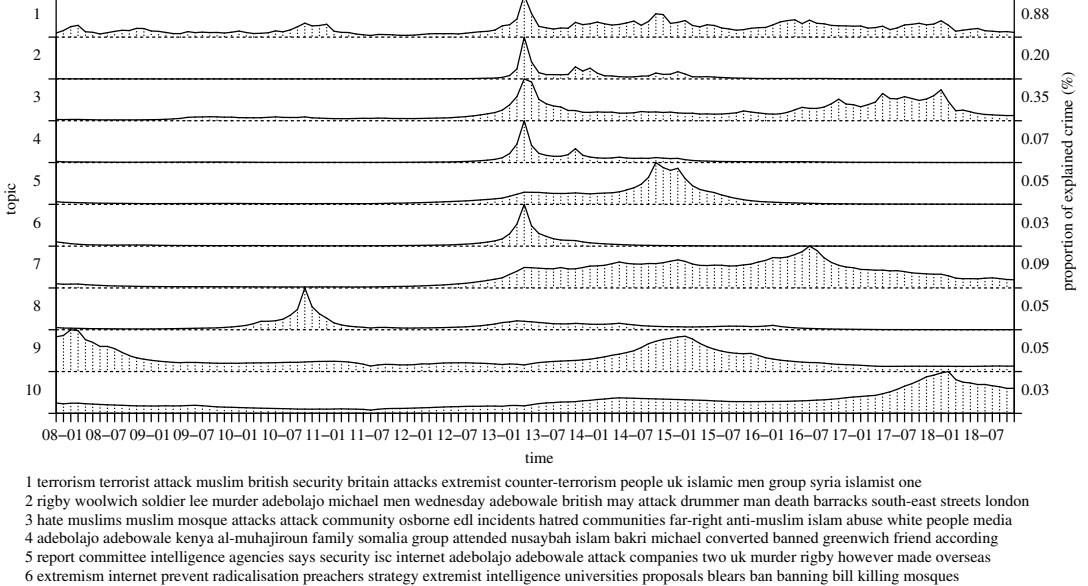

1 terrorism terrorist attack muslim british security britain attacks extremist counter-terrorism people uk islamic men group syria islamist one
2 rigby woolwich soldier lee murder adebolajo michael men wednesday adebowale british may attack drummer man death barracks south-east streets london
3 hate muslims muslim mosque attacks attack community osborne edl incidents hatred communities far-right anti-muslim islam abuse white people media
4 adebolajo adebowale kenya al-muhajiroun family somalia group attended nusaybah islam bakri michael converted banned greenwich friend according
5 report committee intelligence agencies says security isc internet adebolajo adebowale attack companies two uk murder rigby however made overseas
6 extremism internet prevent radicalisation preachers strategy extremist intelligence universities proposals blears ban banning bill killing mosques
7 choudary support isis al-muhajiroun islamic muslims law sharia anjem europe rahman caliphate bakri state banned groups hate muhammad organisations
8 choudhry timms awlaki mp al-awlaki internet iraq anwar stephen lectures war east student sermons yemen ok cleric videos constituency computer
9 terrorism incedal munshi terrorist possessing possession material information useful malik computer likely section explosives nassari documents act
10 hassan train surrey parsons iraq iraqi programme making britain burned prevent green sympathy tube referred hussain islamic causing blamed

**Figure 6.** Terror attacks.

pattern theory explains that certain locations with different points-of-interest attract or repel crime. The topic timelines clearly peak during the event and the associated investigations and court actions and then vanish. These topics show that the model is able to distil accurate and detailed information from the rich data collection with a large vocabulary to detect crime. The top words (of the topics) may be used to query related content online for further analysis.

Figure 5 shows topics related to the London riots that took place in August 2011. All the topics concentrate temporally around this particular date. The topics correspond to a series of crime events,

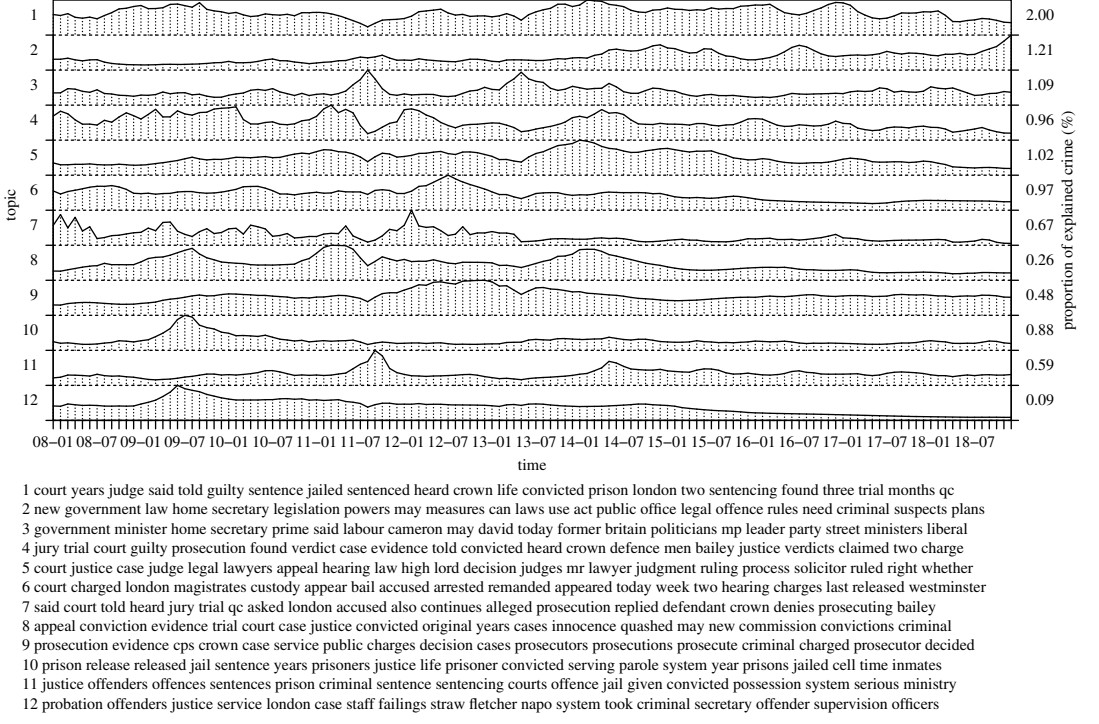

1 court years judge said told guilty sentence jailed sentenced heard crown life convicted prison london two sentencing found three trial months qc
2 new government law home secretary legislation powers may measures can laws use act public office legal offence rules need criminal suspects plans
3 government minister home secretary prime said labour cameron may david today former britain politicians mp leader party street ministers liberal
4 jury trial court guilty prosecution found verdict case evidence told convicted heard crown defence men bailey justice verdicts claimed two charge
5 court justice case judge legal lawyers appeal hearing law high lord decision judges mr lawyer judgment ruling process solicitor ruled right whether
6 court charged london magistrates custody appear bail accused arrested remanded appeared today week two hearing charges last released westminster
7 said court told heard jury trial qc asked london accused also continues alleged prosecution replied defendant crown denies prosecuting bailey
8 appeal conviction evidence trial court case justice convicted original years cases innocence quashed may new commission convictions criminal
9 prosecution evidence cps crown case service public charges decision cases prosecutors prosecutions prosecute criminal charged prosecutor decided
10 prison release released jail sentence years prisoners justice life prisoner convicted serving parole system year prisons jailed cell time inmates
11 justice offenders offences sentences prison criminal sentence sentencing courts offence jail given convicted possession system serious ministry
12 probation offenders justice service london case staff failings straw fletcher napo system took criminal secretary offender supervision officers

**Figure 7.** Court and government actions.

as a result of the shooting of Mark Duggan by police (3.6), and explain violence and rioting (3.1 and 3.4), looting and criminal damage (3.3), stealing (3.7), fires due to arson in specific locations (3.8), riot-related sentences (3.2), police actions (3.5) and riot clean-up (3.9). Again, most of these topics capture unique aspects of violent crime informing police of what type of forces to apply. Topics 3.3 and 3.7 promote the role of youth in committing the offences. The other topics capture three deaths in Birmingham as a result of the spread of rioting in England (3.10), and agitation to riot in social media by certain individuals via spreading messages as well as organizing and executing gatherings (3.11 and 3.12). The latter two topics activate after the riots and provide information regarding post-riot analysis. The topics related to Mark Duggan and Tottenham are active besides the particular dates of riots corresponding to court proceedings and news coverage of the riots as well as overall violence in north London, respectively. The topics express rates of spread and decay at varying speed for crime similarly to the near-repeat crime pattern theory assuming that crime attracts more crime close by both in spatially and temporally. Interestingly, the topics 3.2 and 3.12 that capture riot-related sentences peak quickly after the riots. Because of the bursty nature of riot-related crime, the overall contribution of explaining crime is low but it concentrates in a short time window. The riots provide an opportunity for committing crimes at a large scale and a decreased perceived risk of getting caught because of the high level of disorder.

Figure 6 collects topics related to terrorism in London. A general terror-related topic (4.1) peaks at particular events and the remaining specific attack-related topics have larger proportions at the associated time stamps, as detailed in the following. The topics may inform of overall level of fear in the society, because of the high impact of the offences. The police can use this information to set overall level of alertness and provide resources for the special forces or operations required. The topics cover hate crime (4.3) including mosque attacks and a series of attacks in 2017–2018 with distinct but small peaks and radicalization (4.6) related to the Rigby shooting (4.2), that has major after-effects. Topics 4.4 and 4.5 provide more details of the Rigby attack based on investigations and court proceedings. Other topics cover terror activity of Choudary (4.7) peaking at his court sentence and becoming more active after the Rigby attack and the stabbing of Timms by Choudhry (4.8). Topic 4.9 involves possession of terror-related material. Topic 4.10 captures 2017 Parsons Green tube station bomb attack. However, the temporal pattern suggests a weak association. Overall, the planned and targeted attacks have minor contribution to violent crime counts but are of very high relevance for the society. They are easily masked or confounded by the larger number of more common (either opportunistic or random) violent

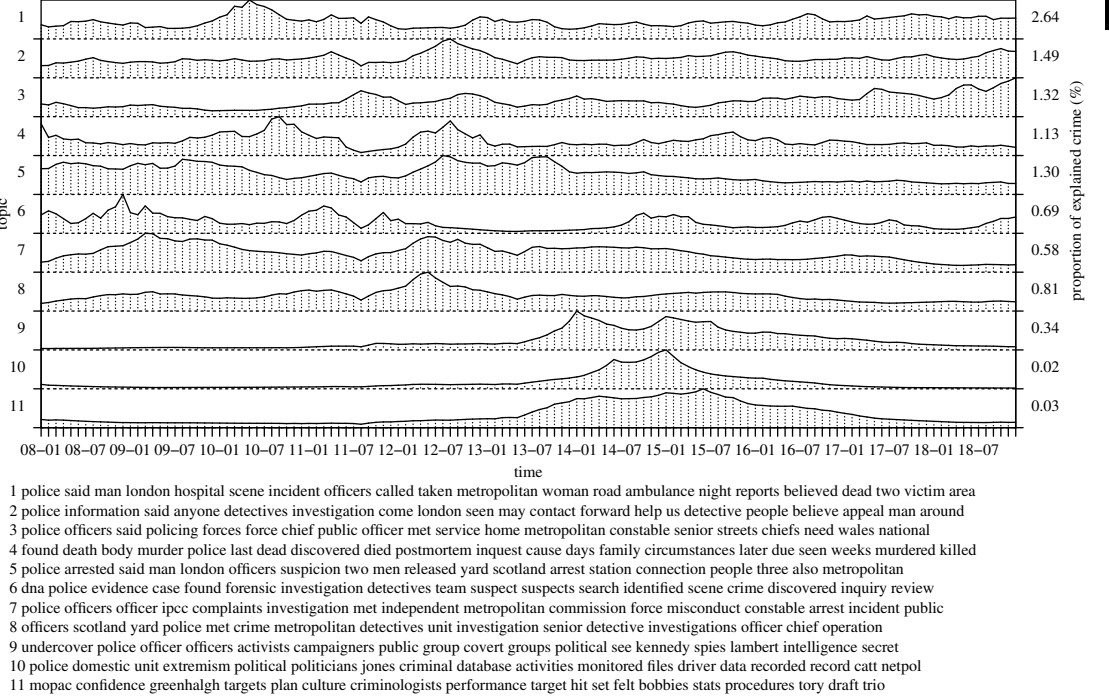

1 police said man london hospital scene incident officers called taken metropolitan woman road ambulance night reports believed dead two victim area
2 police information said anyone detectives investigation come london seen may contact forward help us detective people believe appeal man around
3 police officers said policing forces force chief public officer met service home metropolitan constable senior streets chiefs need wales national
4 found death body murder police last dead discovered died postmortem inquest cause days family circumstances later due seen weeks murdered killed
5 police arrested said man london officers suspicion two men released yard scotland arrest station connection people three also metropolitan
6 dna police evidence case found forensic investigation detectives team suspect suspects search identified scene crime discovered inquiry review
7 police officers officer ipcc complaints investigation met independent metropolitan commission force misconduct constable arrest incident public
8 officers scotland yard police met crime metropolitan detectives unit investigation senior detective investigations officer chief operation
9 undercover police officer officers activists campaigners public group covert groups political see kennedy spies lambert intelligence secret
10 police domestic unit extremism political politicians jones criminal database activities monitored files driver data recorded record catt netpol
11 mopac confidence greenhalgh targets plan culture criminologists performance target hit set felt bobbies stats procedures tory draft trio

**Figure 8.** Police actions.

assaults. Detecting relevant individual attacks based only on the crime counts is not possible but our joint approach is able to uncover these specific incidents. Importantly, some of the attack-specific topics are active after the events corresponding to future crime related to or as a result of these events, prominently related to the hate crime and extremism/radicalization topics.

Figure 7 shows topics related to criminal justice such as court activity (including, sentences, prosecution, charges, trials and appeals), legislation and government statements or interventions. The topics are caused by the offences, although the times between the dates of crime occurrence and sentence often differ. The topics are naturally connected to each other explaining the complex temporal dependencies and fluctuations. We note that the topic of government statements (5.3) coincides with the London riots and some of the terror attacks, showing the need for government intervention and gravity of these events. Also, topic (5.2) captures new legislation as a response to the rise of violent crime or the terror attacks, for instance. Overall, these topics relate to deterring crime and keeping order, explaining their relevance for crime news for the public.

Figure 8 captures topics related to police actions and reports including violent incidents (6.1 and 6.4), investigations (6.2, 6.6 and 6.8) and arrests (6.5). Naturally, these topics are directly associated with violent crime and may capture first available information of the crime. For example, topic 6.2 shows police publicly asking for witnesses. Topic 6.3 involves policing and strength of police forces showing temporal association with the recent rise of violent crime. Topic 6.7 involves complaints against police, partially showing tensions between police and local communities. Lastly, more specific topics capture undercover police operations (6.9) and creation of special police units (6.10 and 6.11). More detailed analysis of the topics may prove useful for assessing policing effects relevant for future decision/policy making.

Finally, figure 9 captures topics related to reasons or theories explaining violent crime. Three prominent topics involve police funding cuts (7.1), youth deprivation (7.2) as well as gangs and youth violence (7.3). Interestingly, these topics clearly co-occur with the London riots and the recent rise of violent crime, figures 5 and 3, respectively. Naturally, reductions in police forces and budget may explain increase of crime via the crime routine theory that promotes the absence of a capable guardian to prevent crime [19] and rational choice theory [20]. In more detail, the latter two topics include concepts of ethnicity, poverty, unemployment, youth, drugs, gang culture, education, weapons, social exclusion and lone parent households. Several studies have established connections between crime and these concepts. See, for instance, Shaw & McKay [21], Sampson *et al.* [22], Sampson & Groves [23], Chamberlain & Hipp [24], Andresen [25], Schreck *et al.* [26] and Hipp &

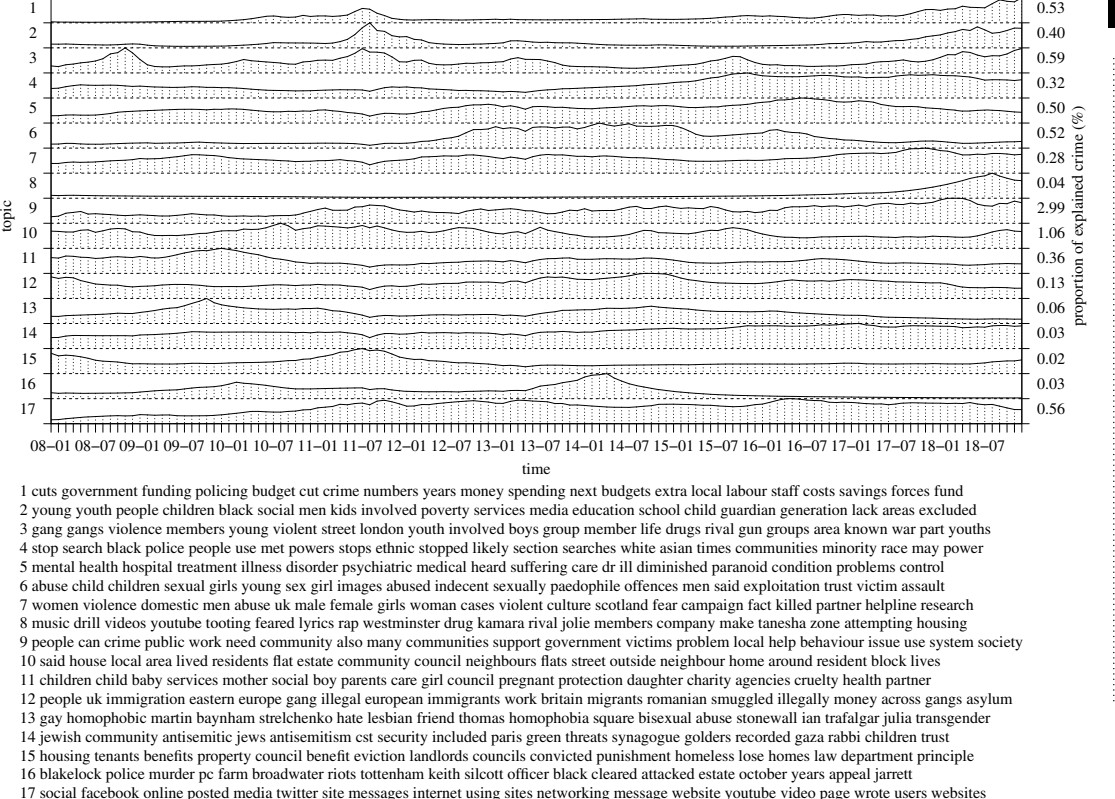

1 cuts government funding policing budget cut crime numbers years money spending next budgets extra local labour staff costs savings forces fund
2 young youth people children black social men kids involved poverty services media education school child guardian generation lack areas excluded
3 gang gangs violence members young violent street london youth involved boys group member life drugs rival gun groups area known war part youths
4 stop search black police people use met powers stops ethnic stopped likely section searches white asian times communities minority race may power
5 mental health hospital treatment illness disorder psychiatric medical heard suffering care dr ill diminished paranoid condition problems control
6 abuse child children sexual girls young sex girl images abused indecent sexually paedophile offences men said exploitation trust victim assault
7 women violence domestic men abuse uk male female girls woman cases violent culture scotland fear campaign fact killed partner helpline research
8 music drill videos youtube tooting feared lyrics rap westminster drug kamara rival jolie members company make tanesha zone attempting housing
9 people can crime public work need community also many communities support government victims problem local help behaviour issue use system society
10 said house local area lived residents flat estate community council neighbours flats street outside neighbour home around resident block lives
11 children child baby services mother social boy parents care girl council pregnant protection daughter charity agencies cruelty health partner
12 people uk immigration eastern europe gang illegal european immigrants work britain migrants romanian smuggled illegally money across gangs asylum
13 gay homophobic martin baynham strelchenko hate lesbian friend thomas homophobia square bisexual abuse stonewall ian trafalgar julia transgender
14 jewish community antisemitic jews antisemitism cst security included paris green threats synagogue golders recorded gaza rabbi children trust
15 housing tenants benefits property council benefit eviction landlords councils convicted punishment homeless lose homes law department principle
16 blakelock police murder pc farm broadwater riots tottenham keith silcott officer black cleared attacked estate october years appeal jarrett
17 social facebook online posted media twitter site messages internet using sites networking message website youtube video page wrote users websites

**Figure 9.** Theories and explanations of violent crime.

Yates [27]. Other topics cover the use of stop and search by police for certain communities (7.4), mental health (7.5) and sexual abuse (7.6) that could underlie the rise of violent crime. Although the effect of the stop and search on crime remains questionable [28], it may underpin tensions between communities and police that may explain crime and disorder. The role of mental health for crime may appear via substance abuse [29] or as a cause for radicalization or criminalization. Currie & Tekin [30] and Widom [31] show that early abuse increases the risk of victims to criminalize later. Of course, abuse, as such, is another type of violent crime. A topic of domestic violence (7.7), associated with abuse, shows a constant temporal trend and may help in detecting violent offenders in the presence of crime under-reporting. Topic 6.8 explains exposure to violence via a particular type of social media content. It is interesting to note that topic 6.16, that captures the aftermath of Broadwater Farm riots in 1985, emerges before and stays active for a long time after the London riots. A more general topic of social media (6.17) becomes more active after the riots, where they played a key role. The remaining topics do not have as clear temporal fluctuations but cover issues around communities (6.9), council housing (6.10 and 6.15), child maltreatment and parenting (6.11), immigration (6.12) and racism (6.13 and 6.14). Here, council housing and social care may reflect regional poverty that is associated with crime to some extent.

# 5. Discussion

Generalized linear models, in particular Poisson regression, are frequently used in crime analysis using points-of-interest or socio-demographics-based regional spatial covariates [32–38]. These models build on environmental criminology addressing geographical profiling or risk surfaces uncovering which spatial covariates are associated with crime [17,39]. Based on these analyses, main established associations between violent crime and the covariates often include deprivation, poverty, ethnicity, lone parent households, unemployment, poor health, low education and high proportion of youth population. We note that these associations also emerge in our results. However, temporal as opposed to spatial

covariates have not been previously used, as considered in this work. Our approach complements well-established spatial crime analysis by incorporating also non-trivial temporal information.

Poisson regression is closely related to matrix factorization. The difference is that for the regression the set of latent variables is fixed and not inferred based on the observed data. Matrix factorization models may be interpreted in a similar well-established manner to regression models. We motivate our model based on this connection. Other non-factorial crime models build on self-exciting point processes [40,41], kernel density estimation [42], Poisson point and Gaussian processes, [43–47], respectively. Importantly, non-factorial crime models are more difficult to interpret and may not be easily extended to provide joint models of crime counts and text. Dynamic matrix factorization models are suitable for temporal crime counts and can account for the clustering property of crime.

Based on the quantitative model comparison, we show that the proposed joint model outperforms a two-step approach that uses text-based covariates inferred based on the text data alone for separate Poisson regression of the crime counts. Unsupervised (dynamic) topic models based on text data alone are not able to uncover topics that are associated with and predictive of crime counts. We also show that predictive performance for our model equals that of a dynamic Poisson matrix factorization model based only on the crime counts. However, the count-based model is less interpretable; the latent variables are not as such associated with topics or text data and the corresponding dynamics/timelines do not correspond to any evident trends or events. Importantly, the added value for our model is improved interpretability.

In the context of crime-related text data, non-dynamic topic models have been used for (moderated) police reports across several crime categories [48] or a single category [49], with the goal to uncover more useful thematic topics to complement ambiguous high-level category-information provided by the police, filtered social media content [50] and short news text snippets, such as breaking news, [51]. Our model differs from these models by incorporating dynamics explicitly and inferring topics that explain and are associated with actual crime counts.

The topic quality naturally depends on the quality of the text data; in this study, we show that the topics are meaningful, easy to interpret and cover an unprecedented range and diversity of themes. Also, the dynamic topic proportions (topic timelines) evidence high quality of the data showing clear and meaningful trends and events.

Gerber [50] uses text data based on social media (location-based tweets) to associate regions with topics. Wang et al. [51] use news tweets (short snippets of breaking news) to infer topics and their temporal proportions. They use the proportions as fixed covariates for crime regression modelling. These works rely on user-generated social media content. The tweets may be noisy and short, presenting some issues for analysis. On the other hand, news articles are provided by professional news agencies and are usually detailed and extensive. We expect the topic quality to suffer for more unstructured and less formal text collections based on user-generated content such as opinions, comments or other social media content.

The developed methodology may be useful for police as such or by complementing or replacing the news articles with sensitive text data provided and accessed by the police officers. It is evident that not all police reports or offences get media coverage and many one-to-one correspondences with actual articles and counts are unobserved. The topics reveal crime trends that may help officers to detect and prevent crime, understand policing demand and focus resources. Naturally, the topic timelines of our model provide important additional information indicating when and what type of resources are needed. On a general level, our model may be used to identify and prevent potential pathways to criminal behaviour including activity in criminal gangs possibly resulting from abuse or mental health issues. The topics may aid in planning for the type and need of proactive support for potential offenders and victims of crime by local councils and relevant institutions. In addition, our model may be used to assess policing effects key for successful policy/decision making. Some of the inferred topics capture long-term trends that provide important insights for proactive policing for future violence and crime overall. More generally, the topics may reveal higher-level content that is of interest to government, legislation, criminologists and society. We expect our results to motivate the police to collect more (natural) text data related to offences in a systematic manner and complement the existing counts with such text data.

The temporal partition of this study is dictated by the available data. To further improve utility of the model, we expect more fine-grained partition to be more useful for crime prevention, focusing on short-term prediction. Also, the specified time range of the study is dictated by the available data. Consistent crime records over a longer time period for crime would provide more up-to-date results but combining crime records over varying crime category definitions is not trivial.

The developed methodology may be useful for other applications. The model may be used for focused analysis of other crime categories such as burglaries and robberies with suitable and available text data. For these crime categories, data confidentiality may be a less severe problem and textual reports by police may be more accessible. Alternatively, we expect the model to be useful for analysis of the global pandemic caused by the COVID-19 by jointly modelling cases/deaths (that are naturally represented as dynamic counts over regions/countries) and related extensive news content.

Ethics. I comply with the Royal Society publishing ethics policy.

Data accessibility. Data and methods are provided in the electronic supplementary material file.

Competing interests. I declare I have no competing interests.

Funding. I received no funding for this study.

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
