## [Peer Review File · Royal Society Open Science]

Review History

RSOS-210750.R0 (Original submission)

Review form: Reviewer 1

Is the manuscript scientifically sound in its present form?

Yes

Are the interpretations and conclusions justified by the results?

Yes

Is the language acceptable?

Yes

Do you have any ethical concerns with this paper?

No

Have you any concerns about statistical analyses in this paper?

No

Recommendation?

Major revision is needed (please make suggestions in comments)

Comments to the Author(s)

The authors introduce an interesting topic model that jointly considers text from news articles on violent crime along with monthly counts of violence against persons crime reports in London. The dynamic topic model uses a nonparametric Bayesian approach with a dirichlet prior, where the novelty arises through a latent variable that is shared by the poisson crime counts and the words in the news reports. The authors compare the approach through a quantitative assessment against a matrix factorization and poisson regression model. They also qualitatively analyze the dynamic topics.

Comments:

1. This paragraph makes little sense to me: "The MF approach is a special case of our model that uses only crime counts. The PR approach uses covariates that consist of the topic proportions of the dynamic topic model, that is a special case of our model that uses only crime articles." Is the reverse true? I would have thought poisson regression only uses crime counts and MF only uses news articles. It would be helpful to mathematically define these sub-models.
2. There are a few things in the quantitative assessment section that I would consider doing differently. Since this is essentially a forecasting (nowcasting?) task, the normal CV split would be in time rather than random. So train the models up to time t_1 and then evaluate on $[t_1, t_{final}]$. It would also be useful to see the predicted counts for each method and how they compare to ground truth. Finally, I would also report AIC/BIC and maybe MAE along with the likelihoods.
3. Why is terrorism showing up in news articles about crime? And is this degrading the performance of the model, since events are being counted that really are different from violence against a person? More explanation/evaluation needs to consider this aspect of the analysis.

Decision letter (RSOS-210750.R0)

Dear Dr Virtanen

The Editors assigned to your paper RSOS-210750 "Uncovering dynamic textual topics that explain crime" have now received comments from reviewers and would like you to revise the paper in accordance with the reviewer comments and any comments from the Editors. Please note this decision does not guarantee eventual acceptance.

Please submit your revised manuscript and required files (see below) no later than 21 days from today's (ie 11-Aug-2021) date. Note: the ScholarOne system will 'lock' if submission of the revision is attempted 21 or more days after the deadline. If you do not think you will be able to meet this deadline please contact the editorial office immediately.

on behalf of Professor Mark Girolami (Associate Editor) and Marta Kwiatkowska (Subject Editor)
openscience@royalsociety.org

Reviewer comments to Author:

Reviewer: 1

Comments to the Author(s)

The authors introduce an interesting topic model that jointly considers text from news articles on violent crime along with monthly counts of violence against persons crime reports in London.

The dynamic topic model uses a nonparametric Bayesian approach with a dirichlet prior, where the novelty arises through a latent variable that is shared by the poisson crime counts and the words in the news reports. The authors compare the approach through a quantitative assessment against a matrix factorization and poisson regression model. They also qualitatively analyze the dynamic topics.

Comments:

1. This paragraph makes little sense to me: "The MF approach is a special case of our model that uses only crime counts. The PR approach uses covariates that consist of the topic proportions of the dynamic topic model, that is a special case of our model that uses only crime articles." Is the reverse true? I would have thought poisson regression only uses crime counts and MF only uses news articles. It would be helpful to mathematically define these sub-models.

2. There are a few things in the quantitative assessment section that I would consider doing differently. Since this is essentially a forecasting (nowcasting?) task, the normal CV split would be in time rather than random. So train the models up to time t_1 and then evaluate on $[t_1, t_{final}]$. It would also be useful to see the predicted counts for each method and how they compare to ground truth. Finally, I would also report AIC/BIC and maybe MAE along with the likelihoods.

3. Why is terrorism showing up in news articles about crime? And is this degrading the performance of the model, since events are being counted that really are different from violence against a person? More explanation/evaluation needs to consider this aspect of the analysis.

===PREPARING YOUR MANUSCRIPT===

===PREPARING YOUR REVISION IN SCHOLARONE===

- An individual file of each figure (EPS or print-quality PDF preferred [either format should be produced directly from original creation package], or original software format).
- An editable file of each table (.doc, .docx, .xls, .xlsx, or .csv).
- An editable file of all figure and table captions.

- Any electronic supplementary material (ESM).
- If you are requesting a discretionary waiver for the article processing charge, the waiver form must be included at this step.
- If you are providing image files for potential cover images, please upload these at this step, and inform the editorial office you have done so. You must hold the copyright to any image provided.
- A copy of your point-by-point response to referees and Editors. This will expedite the preparation of your proof.

- Ensure that your data access statement meets the requirements at <https://royalsociety.org/journals/authors/author-guidelines/#data>. You should ensure that you cite the dataset in your reference list. If you have deposited data etc in the Dryad repository, please include both the 'For publication' link and 'For review' link at this stage.
- If you are requesting an article processing charge waiver, you must select the relevant waiver option (if requesting a discretionary waiver, the form should have been uploaded at Step 3 'File upload' above).
- If you have uploaded ESM files, please ensure you follow the guidance at <https://royalsociety.org/journals/authors/author-guidelines/#supplementary-material> to include a suitable title and informative caption. An example of appropriate titling and captioning may be found at https://figshare.com/articles/Table_S2_from_Is_there_a_trade-off_between_peak_performance_and_performance_breadth_across_temperatures_for_aerobic_scope_in_teleost_fishes_/3843624.

Author's Response to Decision Letter for (RSOS-210750.R0)

See Appendix A.

Decision letter (RSOS-210750.R1)

Dear Dr Virtanen,

It is a pleasure to accept your manuscript entitled "Uncovering dynamic textual topics that explain crime" in its current form for publication in Royal Society Open Science. The comments of the reviewer(s) who reviewed your manuscript are included at the foot of this letter.

on behalf of Professor Mark Girolami (Associate Editor) and Marta Kwiatkowska (Subject Editor)
openscience@royalsociety.org

Appendix A

Response to the Reviewer Comments

We would like to thank the reviewer for useful comments. Please find in the following our responses to these comments.

“1. This paragraph makes little sense to me: "The MF approach is a special case of our model that uses only crime counts. The PR approach uses covariates that consist of the topic proportions of the dynamic topic model, that is a special case of our model that uses only crime articles." Is the reverse true? I would have thought poisson regression only uses crime counts and MF only uses news articles. It would be helpful to mathematically define these sub-models.”

Response: MF refers to a matrix factorisation model that is used to explain the crime counts over different regions and time stamps. The MF model provides a decomposition of the crime count matrix and we can use the model for evaluating predictive accuracy. The model is explained in more detail in Section 3 and Figure 1. Based on Figure 1, the MF model contains only α_t , y_t and ω . All the relevant model details/equations are given in Section 3. The MF model does not use crime news articles in contrast with our joint model of both crime counts and news articles. Hence, the MF model may be interpreted as a special case of our joint model that uses only crime counts.

The PR approach refers to Poisson regression model for the crime counts. Here, the covariates or inputs of the regression model correspond to inferred latent variables of a dynamic text-based topic model. Based on Figure 1, this dynamic topic model contains nodes below the left edge of α_t . Importantly, the dynamic topic model uses only news articles and does not use crime counts y_t to infer the topics and latent variables α_t . The PR model is otherwise similar to the MF model with the distinction that the latent variables α_t inferred by the dynamic text-based topic model are fixed and not updated. In contrast, our joint model is able to update the latent variables α_t by taking into account both crime counts and news articles.

In the revision on page 8, we have accordingly clarified the comparison models, as suggested. We also include mathematical definitions for the comparison models, as suggested.

“2. There are a few things in the quantitative assessment section that I would consider doing differently. Since this is essentially a forecasting (nowcasting?) task, the normal CV split would be in time rather than random. So train the models up to time t_1 and then evaluate on $[t_1, t_{final}]$. It would also be useful to see the predicted counts for each method and how they compare to ground truth. Finally, I would also report AIC/BIC and maybe MAE along with the likelihoods.”

Response: In the quantitative assessment, our main motivation is to evaluate overall the utility of jointly modelling crime news articles and counts. We acknowledge that nowcasting, as suggested, is a relevant and related task, but the setting of nowcasting promotes information that is relevant mainly for left-out time stamps biasing overall evaluation. For this reason, we resort to randomly sampling held-out

crime counts. Future work is needed to fully evaluate the prediction-oriented nowcasting task.

As suggested, we have included in the revision an image (Figure 2) of actual counts and predictions over time stamps for all methods, for a particular fold/mask and three regions, to prevent visual clutter. We note that, the predictions look similar for the remaining areas. From the figure, we see that our approach and MF approach perform equally well and much better than the PR approach. We have extended the revised text (page 8) accordingly to discuss these results.

As suggested, we have included Watanabe-Akaike Information Criteria (WAIC) value that is an alternative for AIC/BIC. We note that WAIC is more appropriate than AIC/BIC for large-scale Bayesian latent variable models that may have a large number of unknown variables. Also, MAE values are included. In the revision, we introduce these measures on page 7. The added values in Tables 1 and 2 further support our conclusions of the model selection and comparison, respectively.

“3. Why is terrorism showing up in news articles about crime? And is this degrading the performance of the model, since events are being counted that really are different from violence against a person? More explanation/evaluation needs to consider this aspect of the analysis.”

Response: The data collection is explained in Section 2. The source of the news articles, The Guardian, associates terror-related news articles with crime tags and hence they appear in our data collection. We note that crime events/counts are counted based on the UK Police data source. Accordingly, only events that are categorised by the Police as violence against a person are included in our data collection. As detailed in the main text, the terror-related topics may capture crime counts related to hate crime and radicalisation as well as overall temporary large-scale disorder that are categorised as violent crime. We also wish to emphasise that our model is able to account for or explain away text content that is not related to crime counts.